# Drought Stress Responses: Coping Strategy and Resistance

**DOI:** 10.3390/plants11070922

**Published:** 2022-03-29

**Authors:** Hanna Bandurska

**Affiliations:** Department of Plant Physiology, Poznan University of Life Sciences, Wołyńska 35, 60-637 Poznań, Poland; hanna.bandurska@up.poznan.pl

**Keywords:** drought, state of stress, tolerance, avoidance, yield, stress survival

## Abstract

Plants’ resistance to stress factors is a complex trait that is a result of changes at the molecular, metabolic, and physiological levels. The plant resistance strategy means the ability to survive, recover, and reproduce under adverse conditions. Harmful environmental factors affect the state of stress in plant tissues, which creates a signal triggering metabolic events responsible for resistance, including avoidance and/or tolerance mechanisms. Unfortunately, the term ‘stress resistance’ is often used in the literature interchangeably with ‘stress tolerance’. This paper highlights the differences between the terms ‘stress tolerance’ and ‘stress resistance’, based on the results of experiments focused on plants’ responses to drought. The ability to avoid or tolerate dehydration is crucial in the resistance to drought at cellular and tissue levels (*biological resistance*). However, it is not necessarily crucial in crop resistance to drought if we take into account agronomic criteria (*agricultural resistance*). For the plant user (farmer, grower), resistance to stress means not only the ability to cope with a stress factor, but also the achievement of a stable yield and good quality. Therefore, it is important to recognize both particular plant coping strategies (stress avoidance, stress tolerance) and their influence on the resistance, assessed using well-defined criteria.

## 1. Introduction

Unfavorable environmental conditions frequently affect plants’ performance, both in natural and agricultural settings. Cramer et al. [1] reported that only 3.5% of the global land area is free from any environmental constraints. Therefore, plants are often exposed to abiotic stress factors which affect their proper development and limit crop production [2,3,4]. Being immobile organisms, plants have been forced to develop specific adaptive traits and ability to adjust (acclimate) to adverse conditions (Figure 1). Adaptation includes developmental, morphological, and physiological traits which help the growth under adverse conditions. Acclimation (hardening) comprises structural, physiological, and biochemical changes responsible for the adjustment to new environmental conditions. It should be distinguished from adaptation, which usually refers to evolutionarily created and genetically determined traits. The ability to acclimate is determined by plant plasticity and includes activation of several complex cellular and molecular responses such as changes in hormone balance and gene expression [5].

The number of papers that focus on the mechanisms of plants response and resistance to stress factors has increased several folds since the beginning of this century [1,6]. It should be highlighted that the understanding of resistance to stress differs depending on the plant’s strategy and plant user’s expectations. Therefore, it is very important to correctly define resistance to stress using clear and appropriate criteria. Stress resistance should not be confused with stress tolerance, which often happens and leads to some misunderstanding which, in my opinion, makes it difficult to define the traits involved in crop resistance to environmental limits. Scientific research focused on identifying the mechanisms or traits responsible for the resistance of crops to stress factors should consider the differences between stress tolerance and stress resistance. The environmental constraint that most often causes the loss of yield throughout the world is drought [2,7]. This paper discusses plant strategies responsible for coping with drought stress and the involvement of the components of these strategies in crop resistance to drought.

## 2. Concept of Stress and Terminology of Stress Resistance

A stress factor affects the state of stress (strain) in plant cells, which leads to structural or metabolic dysfunctions (growth inhibition, damage of structural and functional proteins, inhibition of enzyme activity) and death, or triggers changes that help the plant to adjust to adverse conditions. The plant response depends on the duration and severity of the stress factor, as well as on genetic traits that determine the ability to cope with stress. Depending on the level of stress and duration, plants can experience the state of eustress or distress (Figure 2). A low dose of stressor causes a slight strain (eustress), which triggers responses that help to cope with harmful conditions. Distress caused by a high dose of stressor rapidly triggers the state of stress in plants, leading to physiological destabilization and death or activation responses that protect against stress damage [8]. Plants’ resistance to stress resulting from either adaptation or acclimation may be the effect of activation of diverse coping strategies including stressor escape, stress avoidance (avoidance of the state of stress in cells), and stress tolerance (tolerance of the state of stress in cells). The strategy of stressor escape (adaptive strategy) relies on the adjustment of the life cycle to the period when plants’ needs are met. It can be observed in drought-sensitive plant species, growing in arid and semi-arid areas with regular water deficit, as well as in early spring plants living in a temperate climate. These plants start to develop at the end of winter (February/March) and complete their life cycle at the beginning of spring. Such a strategy is also observed in perennial plants of a temperate climate, which become dormant at the end of autumn to avoid low winter temperatures. The process of preparing plants to survive winter is autumn leaf senescence, controlled by environmental conditions (light, temperature) which affect the relocation of nitrogen, phosphorus, and other elements from leaves to other organs as well as increased levels of endogenous ABA, responsible for dormancy [9]. Maintaining seed dormancy under harsh conditions, regulated by the interplay between ABA and gibberellins, is also considered a stress escape strategy [10]. Stress avoidance is based on the traits and modifications that prevent the occurrence of the state of stress in plant cells, through retardation or weakening of the action of the stressor—as in, for example, stomatal closure responsible for the restriction of water loss through leaves, as well as osmotic adjustments in plant growing under water deficit conditions [2,11]. Stress tolerance, on the other hand, includes mechanisms responsible for coping with the ongoing state of stress in plant cells, such as the synthesis of compatible substances and proteins, which protects against the negative effect of osmotic and ionic stresses in drought- and salt-stressed plants [11,12]. In other words, it is the capacity to sustain plant functions, thanks to the modifications that counter negative effects of the occurrence of the state of stress, and to repair the damage after stress relief (Figure 2).

Both stress avoidance and stress tolerance are responsible for resistance to stress, understood as the ability to cope with adverse environmental conditions, by keeping a balance between growth, reproduction, and activation of suitable coping strategies [13]. This kind of resistance can be called *biological resistance*, which is the strategy of an individual plant to tolerate and survive stress conditions. An example of *biological resistance* is also the stressor escape strategy which occurs in stress-sensitive plants. From the perspective of plant users, crop resistance to environmental stresses should be defined as the ability to cope with stress conditions thanks to defense responses (stress tolerance and/or avoidance) which enables maintenance of stable and good quality yields. Therefore, it can be called *agricultural resistance*.

## 3. Plant Responses to Drought

Drought is a meteorological term defined as a period of little or no rainfall, which reduces the amount of water in the soil, and is usually accompanied by high evaporative demand, leading to continuous loss of water by transpiration. It is considered the most frequent climate-related constraint in many regions of the world [3,14]. This stress factor generates the state of stress (strain) in plant cells, which is the reduction in water content (dehydration, water deficit), adversely affecting plant physiological activity, growth, reproduction, and crop productivity [15]. The level of dehydration depends on stress severity and duration, as well as on adaptive traits protecting against water loss (smaller leaves, leaves covered with cuticle or tomentose, as well as leaf folding) and supporting water uptake from deeper soil layers (extensive vertically orientated root system). Another example of an adaptive trait protecting from water loss is stomatal behavior (stomata open at night and closed during day) in crassulacean acid plants (CAM) having an alternative route of carbon assimilation which occurs during the night [5,16]. The above-mentioned adaptations do not usually occur in crop plants, which mostly belong to mesophytes, and are able to grow in an environment with a moderate supply of water. These plants may adjust to water scarcity through the activation of stress avoidance and/or tolerance mechanisms directed at preventing dehydration and/or dehydration damage, and surviving stress [11,13]. A suitable and commonly used marker to evaluate the level of dehydration (state of stress) is relative water content (RWC). In leaves of well-irrigated plants RWC is ≥90%, but with mild drought stress it is in the range 60–70%, with moderate stress 40–60%, and in the case of severe stress it is lower than 40% [17].

Based on the ability to maintain stable leaf hydration under water deficit conditions the water management strategy of plants is classified as isohydric or anisohydric [18]. Isohydric species (‘water savers’) maintain nearly constant RWC through precise control of stomatal behavior. These plants respond to drought by a rapid decrease in stomatal conductance (g_s_) and restriction of excessive water loss without a reduction in leaf area but at the same time show a decrease in photosynthetic activity. In contrast, anisohydric plants (‘water wasters’) show a decrease in leaf water content and strong leaf area reduction but keep stomata open and maintain a high photosynthetic rate [19,20,21]. The extent of tissue dehydration is a signal triggering, directly or through ABA increase, the activation of appropriate metabolic and physiological changes responsible for plants’ adjustment to drought [22]. Even a slight decrease in RWC triggers upstream signaling events, leading to ABA accumulation and stomatal closure [23]. It was suggested that ABA is primarily synthesized in roots, then it is moved to shoots via xylem vessels and acts as a signal of soil water shortage [24]. Recent research revealed that the source of ABA accumulation in roots under drought stress conditions is its transport from leaves [25,26]. In the model plant *Arabidopsis thaliana* it was observed that CLE-25 peptide is a root-derived molecule which moves via the vasculature to leaves and transmits a water deficit signal triggering ABA synthesis by the activation of biosynthetic enzymes [27]. The root–shoot signal mediating the effect of soil water deficit on stomata in tomato comprises a dialogue between ABA and strigolactones, ethylene precursor ACC, or sap pH [28]. Stomatal closure in response to soil water deficit in maize and poplar (isohydric species) is regulated by the interaction between the hydraulic signal as a primary message and ABA as a secondary message [18,28,29].

ABA is also involved in several downstream events responsible for the maintenance of tissue hydration (dehydration avoidance strategy), which include osmotic adjustment, comprising the accumulation of organic osmotic compounds (proline, glycine-betaine, soluble proteins, carbohydrates) in leaves and in roots [15,22]. It appears to be necessary for the activation of proline transport and deposition in the root growing region, allowing the maintenance of root growth and undisturbed water uptake under drought conditions [29]. Indeed, ABA plays a central role in plants’ response to drought at different levels of organization; however, it does not act alone but through synergistic or antagonistic crosstalk with other hormones. Crosstalk between ABA, ethylene (ET), and auxin (AUX) regulates root growth and architecture [30]. Guard cells’ aperture and water loss by stomata are controlled by the orchestration of ABA with jasmonic acid (JA), ET, salicylic acid (SA), as well as AUXs and cytokinins (CKs). Increased levels of JA, ET, and SA in drought-stressed plants promote the induction of stomatal closure which is maintained by the decline in CK and AUX levels [31,32,33,34]. Foliage-derived ABA promotes root growth under drought by lowering the level of ET, which is a root growth inhibitor [35]. Additionally, auxin-induced alteration of root architecture, which leads to the creation of more vertical and deeper roots, plays an important role in maintaining better water acquisition under drought conditions [27,36]. Brassinosteroid (BR)-induced root hydrotropism and accumulation of osmoprotectants (proline, trehalose, raffinose) in roots may also improve water uptake under drought [27,37]. The beneficial effect of ABA on water transport and tissue hydration under drought conditions may also be achieved by its influence on the improvement of root hydraulic conductivity through regulation of the activity of membrane water channels—aquaporins [38,39].

The strategy of dehydration avoidance (isohydric behavior) allows plants to sustain physiological functions under stress conditions and recover after stress termination. This strategy is effective in plants exposed to mild or moderate drought that does not last very long but under prolongated drought it affects carbon starvation [40]. Moreover, when stomata are closed plants absorb more light than can be used in carbon fixation, which triggers generation of reactive oxygen species (ROS), affecting secondary stress and damage of PSII, leading to further weakness of photosynthesis [12,41]. What is more, during long-term drought the ability of plants to maintain stomatal closure may be weakened due to a decrease in ABA level and plant behavior changes to anisohydric [42]. The response to drought in anisohydric plants (barley, wheat, sunflower) is mainly regulated hydraulically. The maintenance of stomatal conductance in these plants is supported by the capacity for osmotic adjustment, controlled by the dehydration signal, which enables plants to extract water from soil to maintain tissue hydration [18,40]. In anisohydric wheat genotypes the level of ABA in leaves did not change under water deficit conditions, while in roots it increased but only after 21 days of stress [29]. Therefore, it is possible that, along with tissue dehydration, ABA may also play a role in the response of anisohydric species to prolonged drought. The stomatal conductance in anisohydric plants is also maintained by undisturbed water movement through cell membrane aquaporins responsible for roots’ ability to conduct water [43]. It was reported that ABA increases the activity of aquaporins and improves root hydraulic conductivity [44]. The activity of aquaporins is also regulated by gibberellins (GAs), CKs, methyl jasmonate (MeJA), and AUXs at transcriptional and post-transcriptional levels ([45] and references therein). The anisohydric strategy is beneficial under mild to moderate drought conditions but may be a risk under severe and long-term stress, which may cause hydraulic failure and severe dehydration [20].

In plants exposed to severe and long-term drought, dehydration cannot be avoided, and activation of dehydration tolerance mechanisms becomes important. Dehydration has a deleterious effect on cell membranes and causes the disruption of many biochemical and physiological processes [2,46]. A frequently used indicator of dehydration tolerance is the cell membrane injury index or membrane stability index, which shows the ability to maintain membrane integrity at a given level of dehydration [47,48]. The dehydration tolerance mechanisms enable plants to maintain membrane integrity and cell homeostasis, and to regain physiological activity after stress cessation [12,41]. These mechanisms are controlled by ABA-dependent and -independent pathways and include synthesis of protective proteins (LEA proteins, dehydrins, chaperons) and compatible compounds (proline, glycine-betaine, proline-betaine, trehalose, raffinose mannitol, pinitol) involved in enzyme and membrane protection [2,22,41,49,50]. Dehydration-induced disturbance of the respiratory metabolic pathway exhibits generation of ROS, leading to a state of oxidative stress [2,46,51,52,53]. Moreover, in drought-stressed plants the enhanced build-up of ROS is caused by photosynthesis disruption and increased photorespiration due to the limitation of CO_2_ uptake [53,54]. Overproduction of ROS (secondary stress), which includes superoxide radicals (O_2_^•−^), hydrogen peroxide (H_2_O_2_), hydroxyl radical (OH^•^), and singlet oxygen (^1^O_2_), is harmful to organelles through lipid peroxidation and damage to nucleic acids and proteins [2,3,46]. In order to overcome oxidative damage, plants possess enzymatic and non-enzymatic ROS-scavenging systems. Enzymatic antioxidants include superoxide dismutase (SOD), catalase (CAT), and peroxidases (POX). The non-enzymatic components of the antioxidative system comprise ascorbic acid, α-tocopherol, flavonoid, glutathione, carotenoids, proline, and phenolic compounds which mitigate oxidative damage by direct reduction of ROS activity and by working together with antioxidant enzymes [53,55]. Additionally, alternative oxidase (AOX) is involved in avoidance of excess generation of ROS in mitochondrial electron transport chains [54]. ABA plays a pivotal role in the activation of antioxidant enzymes and synthesis of low molecular ROS scavengers [49,50]. Upregulation of the antioxidant system may also be controlled by JA, SA, and BRs [34,51,52,53,54,55,56,57,58].

Thanks to the efficient antioxidative system, plants can keep ROS at non-toxic levels, and these molecules are thought to act as signals for activation of stress defense responses [45,54]. It was also evidenced that NADPH oxidase localized in apoplastic fluid is involved in ROS production for integrating signaling networks involved in stress response processes. An increased level of this enzyme was detected in drought-stressed rice as well as in leaves of ABA- and Ca^+^-treated maize seedlings [54,59]. Moreover, NADPH oxidase regulates H_2_O_2_ production for the signaling cascade which affects ABA-dependent stomatal closure and antioxidant defense. The involvement of NADPH oxidase in brassinosteroid-induced H_2_O_2_ production and regulation of stomatal closure/opening and antioxidant defense was also reported [54,60].

Plant responses to drought are governed by a sophisticated regulatory system working at the molecular level. The decrease in turgor pressure leads to tension changes in plasma membranes, which are perceived by membrane proteins including receptor-like kinases (RLKs), histidine kinases (HKs), and integrin-like proteins (ILPs) working as osmotic stress sensors. ATHK1 is an *Arabidopsis thaliana* His kinase postulated to play a role in water stress perception triggering the mitogen-activated protein kinase (MAPK) signaling cascade both in ABA-dependent and ABA-independent regulatory systems [61]. A crucial role in the signal transduction route is played by transcription factors (TFs) that bind to TF binding sites (TFBS) in the promotor region and regulate gene expression. TF families involved in plants’ response to drought include bZip (AREB/ABF), AP2/ERF (DREB/CBF), MYB/MYC, WRKY, and NAC [3,62]. In the ABA-dependent pathway the perception of ABA by receptor proteins is the primary event that triggers downstream signaling cascades to induce final physiological responses. The receptors for this hormone are small soluble cytosol/nucleus-localized pyrabactin resistance (PYR)/PYR-like (PYL)/regulatory components of ABA receptor (RCAR) proteins. The interaction of ABA with PYR/PYL/RCARs affects deactivation of protein phosphatase enzymes (PP2Cs), which are constitutive negative regulators of ABA-induced responses. The inhibition of PP2Cs leads to auto-phosphorylation of the protein kinases SnRK2s, which induces stomatal closure and stimulates nuclear targets that trigger expression of various water stress associated genes due to activation of TFs [62]. ABA-dependent gene expression systems involve activation of b-ZIP (AREBs/ABFs), MYC/MYB, as well as NAC transcription factors [63].

In ABA-independent responses to drought the dehydration signal from the cell surface to the nucleus is mediated by calcium, JA, and ROS [62]. Water deficit leads to membrane destabilization and Ca^2+^ influx into the cytoplasm. The calcium signal is detected and transduced through calmodulin (CaM), calcium dependent protein kinases (CDPK), and calcineurin B-like proteins (CBLs) and interacts with the MAPK cascade, leading to activation of TFs (DREB, NAC) and expression of genes coding the synthesis of functional proteins (LEA proteins, chaperones, dehydrins, enzymes of osmolyte biosynthesis). JA, on the other hand, is engaged in activation of the MYC2 transcription factor, which triggers expression of stress-responsive genes [62]. Furthermore, JA along with ROS acts as a stress-signaling unit triggering the expression of genes involved in activation of enzymatic and non-enzymatic scavenging events [62,64]. The widespread plant response to drought is proline accumulation due to the stimulation of its synthesis from glutamate catalyzed by pyrroline-5-carboxylate synthetase (P5CS) and pyrolino-5-carboxylate reductase (P5CR) [65,66]. Synthesis of this amino acid under drought is driven by both ABA-dependent and ABA-independent signaling pathways engaged in triggering expression of *P5CS* and *P5CR* genes regulated by many TFs, which are also related to responses to drought controlled by other growth regulators [67].

Important components of the stress-factor-induced regulatory system are epigenetic modifications which are independent of DNA sequence changes. These changes include chromatin remodeling such as DNA methylation and histone modifications altering the structure and accessibility of chromatin, leading to changes in gene expression at the transcriptional and post-transcriptional levels [68]. Drought-stress-induced changes in DNA methylation have been observed in diverse plant species. These changes were related to the expression of genes encoding transcription factors and were involved in drought resistance mechanisms or were linked to drought sensitivity [69,70,71]. It was found that changes in DNA methylation (demethylation) in water deficit stressed rice were responsible for proline accumulation via the upregulation of proline metabolism-related gene expression [72]. In addition to DNA methylation, drought-induced histone modifications (methylation, acetylation) are involved in controlling gene expression in stressed plants [73]. It was observed that drought stress triggered histone H3 lisyne4 tri-methylation (H3K4 me3) in the gene body region of nine cis-epoxycarotenoid dioxygenase 3 (NCED3), which is a key enzyme involved in ABA synthesis. Additionally, some studies reported the increase in H3K4me3 and H3 lisyne9 acetylation (H3K9Ac) in the promotor region of such genes as *RD29A*, *RD29B*, *RD22*, and *RELATED TO AP2.4* (*RAP2.4*) encoding synthesis of LEA proteins. The abundance of histone modification and the number of genes expressed depend on stress duration and degree [70,73]. Most of the epigenetic modifications are removed when the stress is relived, but some of them persist, enabling plants to remember past stress and to prepare for future recurrent stress events which occur during plant life. This is so-called “plant stress memory”, which can also be transferred to further generations during sexual and vegetative reproduction [69,71,74]. Integral components of the stress response at the molecular level also involved in memory pathways are non-coding small RNAs (miRNAs, siRNAs), which can trigger DNA methylation and histone modifications. Plants exposed to drought can memorize stress events through DNA and histone modifications for specific gene expression thanks to up- and downregulation of small RNAs responsible for the increased resistance to future stress events through the control of TFs, ROS, and hormone levels [71,74].

## 4. Drought Coping Strategies and Resistance

The ability to avoid or tolerate dehydration is crucial in dealing with drought at cellular and tissue levels (*biological resistance*), which allows plants to survive during water scarcity conditions and recovery. The tolerance and avoidance mechanisms were developed during evolution in order to adjust to environmental conditions but usually do not have beneficial effects in agricultural production. Plants can withstand drought without any visible signs of dehydration and/or dehydration damage, but their growth and yield may be lower than expected. This is an unwanted side effect of plant adjustment to stress, which has a negative impact on biomass accumulation and yield (*agricultural resistance*).

The activation of coping mechanisms is connected to increased energy and nutrient consumption, which results in the allocation of less energy and assimilates to growth processes, leading to yield reduction [15,75]. Furthermore, many traits associated with drought resistance have a dual effect (positive or negative) on plant productivity which depends on stress intensity and timing as well as on climatic conditions such as light intensity and evaporative demand [76]. The dehydration avoidance strategy, such as stomatal closure, reduces water loss from leaves. However, at the same time it causes the restriction of CO_2_ uptake, ROS generation, damage of PSII, and the inhibition of photosynthesis, resulting in the reduction of crop production [12,41,77,78,79]. Moreover, changes in the hormonal balance, which is a part of the coping strategy consisting of an increase in the levels of ABA, JA, Et, and SA, and decrease in CKs, AUXs, and GAs, may also bring about photosynthesis inhibition, growth restriction, leaf senescence acceleration, and leaf fall, negatively affecting yield [32,34,80]. Therefore, there is a conflict between plant coping strategies (avoidance, tolerance) and resistance to drought essential for agricultural production. In the agricultural perspective, drought-resistant plants are those that maintain growth and stable yield during water-limited conditions. The priority in breeding research focusing on improving drought resistance is to obtain crop genotypes that can cope with drought stress without growth and yield reduction. Therefore, the research on plant stress physiology should concentrate on finding those features of coping strategies that ensure growth maintenance and stable yield (Table 1).

Many genes and processes involved in plants’ ability to cope and survive drought (*biological resistance*) in experiments conducted under laboratory conditions have been identified. However, the knowledge about their real function in the resistance to this stress, based on well-defined agronomic criteria (*agricultural resistance*), is rather poor [11,22,81,82]. It is hard to show the involvement of a particular trait or adjustment to drought in maintaining yield potential in a short-term experiment. The response to short-term drought conditions in soil pot experiments (limited rhizosphere) did not reflect the response to long-term water shortage in the field [83]. Drought resistance is the result of combined processes that happen on different timescales and have a long-term impact on plant performance and yield. Short-term responses to drought include triggering physiological feedback processes responsible for stabilizing plant water and carbon status, which are often not correlated with the long-term effect. The favorable effects of these feedback strategies on yield depend on the drought scenario as well as on scalability and phenotypic distances between traits involved in particular coping strategies and those responsible for yield [28]. It is necessary to search for processes and adjustments that allow crops to continue to grow under water-limited conditions and rapid recovery after stress termination without yield reduction.

The source of traits valuable in developing new drought-resilient crop varieties may be wild genotypes and landraces originating from rainfed areas [3]. Another promising approach is the introduction of new crop species able to cope under water-limited conditions and maintain stable growth. An interesting species in this regard is quinoa (*Chenopodium quinoa* Willd.), which originated in the Andean region. It has begun to be called ‘the 21st century crop’, and recently it has been introduced into cultivation in many regions of the world. Quinoa has received special attention due to its high nutritional composition of seeds and strong natural ability to cope with drought [84,85]. There is wide diversity among quinoa genotypes in the traits of drought coping strategy (*biological resistance*) and resistance assessed based on the seed yields (*agricultural resistance*). The drought response mechanisms in quinoa to 8endure water deficits include accelerated root growth, high water-use efficiency (WUE), osmotic adjustment, turgor maintenance, increased synthesis of osmoprotectants such as amino acid proline, and soluble sugars (glucose, trehalose), ABA biosynthesis, antioxidant defense, heat-shock, and LEA protein synthesis [86]. Field studies have shown no significant yield reduction in the Danish quinoa cultivar Titicaca under water deficit conditions [84]. Soil pot experiments revealed that the capacity for growth in a drought-prone environment in ‘Titicaca’ was associated with the increase in WUE due to higher ABA concentration and nutrient content [87]. Recent studies revealed that drought resistance in quinoa var. Red Faro was due to elevated recovery capacities of PSII and PSI photochemical activities after re-watering [88]. There are numerous studies focused on molecular, biochemical, physiological, and morphological responses of varied quinoa genotypes to drought both under laboratory and field conditions. The sequencing of the quinoa genome creates the possibility of using new molecular tools to fully discover regulatory mechanisms involved in drought resistance of various quinoa genotypes [86].

A promising drought resistance strategy for crops is the ability to optimize water use, along with sustained high photosynthetic activity, which is an essential component of plant productivity [89]. It may be achieved by triggering varied metabolic and physiological responses of the dehydration avoidance strategy, which includes the modification of root conductivity and architecture, regulation of stomatal behavior allowing the maintenance of photosynthetic CO_2_ fixation, as well as protection against non-stomatal photosynthesis limitation [90,91,92]. Plants with greater WUE assimilate more carbon per unit of transpired water. These plants are less susceptible to drought as they take less water from the soil and may access this water later in the season when a lack of water has become a limiting factor [79]. The improvement in WUE under water-limited conditions without trade-offs in carbon assimilation was revealed in transgenic tomato with overexpression of the gene encoding ABA biosynthesis enzyme (NCED3) as well as in *Arabidopsis* overexpressing ABA receptors [38,93]. The effect of increased ABA levels in roots and leaves of drought-stressed tomato lines was lower stomatal conductance and greater root conductivity [38]. However, ABA signaling-mediated changes in *Arabidopsis* transgenic lines affected reduced stomatal conductance, which was compensated by increased CO_2_ gradients across stomata, allowing maintenance of a CO_2_ influx [93]. These findings in *Arabidopsis* are being considered for translation to cereal crops to obtain drought-resistant genotypes through improving WUE [94,95]. A suitable criterion to measure WUE is carbon isotope discrimination (Δ^13^C), which is used in breeding programs to select drought-resistant crop genotypes [76]. A significant positive relationship between Δ^13^C and yield was revealed in drought-stressed quinoa cultivars under field conditions [86,96]. However, Tardieu [76] considers that Δ^13^C is a positive trait for yield under severe water deficit conditions but under mild to medium drought the positive traits that optimize yield are high stomatal conductance and growth maintenance.

Multiple biochemical and physiological changes that are components of drought coping strategies were revealed to have a favorable effect on yield (Table 1). Lower yield reduction under drought conditions was observed in a wheat cultivar that exhibited osmotic adjustment resulting from the accumulation of soluble sugars and proline as well as increased activity of enzymatic and non-enzymatic antioxidants. These changes allow for the maintenance of high photosynthetic CO_2_ fixation during drought and rapid recovery after re-watering, which are responsible for the final productivity [97]. Barley genotypes that yielded better under drought conditions exhibited increased expression of 34 genes which are involved in stress signaling, carbon metabolism, control of stomatal closure, proline synthesis, activation of the ROS scavenging system, and protective protein synthesis [82]. Elevated osmotic adjustment, increased expression of dehydrin genes, and a significant increase in alpha-tocopherol, which plays an important protective role for PSII, along with a higher photosynthetic rate, were observed in barley genotype, characterized by a smaller decrease in the performance index under drought stress conditions [98]. A large body of evidence has shown a beneficial role of proline in dealing with drought stress (*biological resistance*). Proline, involved in osmotic adjustment, is a free radical scavenger and acts as a compound that protects enzymes, proteins, and cell membranes against detrimental effects of dehydration and oxidative stress ([66,99] and references therein). It also serves as a carbon and nitrogen reserve after stress relief, and may act as a signaling molecule, able to activate defense responses [100]. Therefore, rapid proline accumulation at the beginning of drought stress may play an essential role in the dehydration avoidance strategy. Its increased level may also protect plants from the detrimental effect of dehydration (dehydration tolerance strategy), and it may be involved in the ability to recover after stress cessation. However, the involvement of this amino acid in the resistance to drought, understood as an adjustment without any negative effects on yield, is still not clear. The possible beneficial effect of greater leaf proline accumulation under drought on *agricultural resistance*, based on grain yield, was found in wheat [101]. Interesting results were obtained by Frimpong et al. [102], who observed that introgression barley lines, harboring a pyrroline-5-carboxylate synthase (*P5cs1*) allele, had markedly higher proline content in spikes and leaves, compared with other genotypes. These lines also showed milder drought symptoms, were able to maintain a high photosynthetic rate under drought, and achieved higher final seed production. Moreover, the barley near-isogenic line *NIL 143*, characterized by higher leaf and root proline content, showed less severe symptoms of drought, higher leaf water content, better stomatal conductance and net CO_2_ assimilation than other genotypes. This barley line also exhibited increased lateral root growth, probably due to high proline accumulation [103]. Considerable evidence obtained previously revealed that drought-stress-induced expression of proline biosynthetic genes is regulated by TFs related to almost all plant hormones [67].

One recently considered approach in attaining crop resistance to drought is focused on better understanding of the role of plant growth regulators (PGRs) in the coping strategy along with the mitigation of the negative effect of drought on productivity and yield. PGRs play an important role in triggering, directly or through specific signal cascades, a wide range of metabolic and physiological responses of plants to drought. Many of these responses, which are components of the drought stress coping strategy, are the result of positive or negative interactions between diverse PGRs [31,32,104]. Broadening knowledge about the impact of drought on the fluctuation of the level of PGRs and about the crosstalk between them in triggering appropriate responses seems to be essential in identifying components of drought coping strategies, which permit undisturbed growth and stable yield. The hormone that plays a key role in the plant response to drought is ABA, commonly called a “stress hormone”. An increased level of this PGR in drought-stressed plants acts as a signal that regulates multiple responses at physiological and biochemical levels [10,50]. It was suggested that the interaction between plant hormones (ABA, AUXs, CKs, and ET) may play an important role in a diverse drought response of sensitive and resistant wheat lines [105]. The resistant wheat line was able to maintain growth and was characterized by lower yield reduction under drought. This line was temporarily anisohydric and closed the stomata only at a higher level of drought which correlated with the repression of ABA synthesis. At the same time, it had the ability to activate defense responses (ROS protection, LEA proteins, and cuticle synthesis) and to trigger expression of photosynthesis genes as well as genes involved in AUXs, CKs and Et metabolism, and signaling. However, the drought-sensitive wheat line was isohydric, had a higher ABA level, closed stomata at the start of stress and began photosynthesis inhibition. Certain recently obtained results of research focused on crosstalk between ABA, CKs, and BRs at physiological and molecular levels seem to be promising in finding drought coping strategies that prevent yield reduction [27,33]. ABA increase and the reduction in CK level under drought lead to a decrease in stomatal aperture and density, as well as accelerated leaf senescence, along with photosynthesis inhibition [32,106]. The manipulation of endogenous CK level and control of CK signaling pathway components in transgenic rice were effective in restoration of stomatal conductivity, reduction in leaf senescence, and amelioration of yield losses [32]. This transgenic rice also displayed increased expression of BR-related genes and repression of JA-related genes [107]. It was reported that BRs trigger the expression of various stress-related genes important in the maintenance of photosynthetic activity, stimulation of the antioxidant system, and accumulation of osmoprotectants [32,107,108]. Furthermore, overexpression of the BR receptor (BRL3) leads to activation the synthesis of osmoprotectants (i.e., proline, trehalose, sucrose) in roots and overcoming growth arrest as well as modulating the root hydrotropic response during drought [27,37,104]. An interesting and promising mechanism leading to drought resistance appears to be the involvement of BRs in the expression of cell wall extension and release of enzymes, which lead to increased cell expansion [108]. The last several years of research have shown that crosstalk between BRs and other hormones is involved in the network of complex regulatory responses to drought, including stress perception and signaling leading to activation of various coping strategies [109]. Master regulators of abiotic stress responses whose expression is controlled by hormonal balance and crosstalk are TFs [110]. Gaining knowledge about the pattern of appropriate hormonal balance and crosstalk as well as identification of stress-responsive TFs and their role in activation of the components of the drought coping strategy without yield mortality is a powerful approach for achieving drought-resistant crop cultivars [3,62,111].

**Table 1 plants-11-00922-t001:** Components of coping strategies and agricultural resistance in crops and model plants.

Plant Species/Genotypes	Stress Imposition Stress Level	Components of Coping Strategy	Agricultural Resistance	References
*Arabidopsis thaliana* transgenic line RCAR10-4	soil pot experiment water withholding 8 weeks severe stress RWC—not performed	increased expression of ABA receptor reduced stomatal conductance with maintenance of carbon assimilation	improvement in WUE and growth	[93]
tomato transgenic line sp12	soil pot experiment water withholding at four- or five-leaf stage 5 days RWC—not performed	overexpression of ABA biosynthesis of gene (*NCED*) increased ABA level in root and leaves reduced stomatal conductance increased root hydraulic conductivity, water status improvement	improvement in WUE without trade-offs in carbon assimilation	[38]
wheat ‘Luhan7′	soil pot experiment, irrigation withheld at tillering and jointing stage 10 days moderate stress RWC 85–89%	osmotic adjustment (proline, sugars) stomata closure activation of antioxidant system	high photosynthetic CO_2_ fixation high drought index and harvest index	[97]
barley *H. vulgare* ‘Martin’ *H. spontaneum* HS41-1	soil pot experiment, water withholding at flowering stage 13 days severe stress RWC—not performed	high expression of signal transduction genes (TFs, CDPK, membrane binding proteins) and functional genes directly involved in coping strategy (stomatal behavior, synthesis of glycine-betaine, proline, antioxidants, dehydrins)	higher chlorophyll content and lower grain yield losses than in genotype without enhanced expression of coping strategy genes	[82]
barley ‘Yousof’ and ‘Morocco’	soil pot experiment water withholding at two weeks seedling stress duration? mild stress RWC ~88%	high level of dehydrin and alpha-tocopherol involved in PSII protection in ‘Yousof’	lower reduction in CO_2_ assimilation rate and performance index in ‘Yousof’	[98]
transgenic rice	soil pot experiment water withholding at pre-anthesis and post-anthesis 6–10 days mild stress RWC ~85%	increased CK synthesis, increased expression of BR related genes and repression of JA-related genes modification of source/sink relationships, a stronger sink capacity	higher grain yield with improved quality (nutrients and starch content)	[107]
wheat ‘Zagros’ and ‘Marvdaht’	soil pot experiment soil moisture at about 50% of field capacity 31 days RWC—not performed	higher ABA and proline accumulation in ‘Zagros’ than ‘Marvdaht’	higher harvest index and lower grain yield reduction in ‘Zagros’ than ‘Marvdaht’	[101]
barley introgression lines with wild allele *p5cs1-* *S42IL-141, S42IL-141*	soil pot experiment reduction in irrigation at booting stage mild stress 15-day RWC ~83%	significantly higher spike and leaf proline level than other line	maintenance of high photosynthetic rate and inherent WUE, high final seed productivity	[102]
barley near-isogenic line with wild allele *P5cs1-* *NIL 143*	rhizoboxes filled with soil soil water content decreased from 40% at the beginning to 6% after 17 days (three-leaf seedling) severe stress RWC ~59%	higher root and shoot proline content than in other genotypes, less severe drought symptoms, better stomatal conductance, higher RWC, enhanced root growth	enhanced net assimilation rate	[103]
quinoa ‘Titicaca’	field experiment soil pot experiment mild to severe RWC—not performed	ABA increase, high WUE	no yield reduction	[84,87]
quinoa 10 varieties	field experiment RWC—not performed	high carbon isotope discrimination	high yield	[96]
wheat drought-tolerant ‘Halberd’ drought-sensitive ‘Cranbrook’	soil pot experiment growth chamber water withholding drought stress at the young microspore stage RWC—not performed	ABA increase, stomatal closure at the start of stress, inhibition of photosynthesis in ‘Halberd’ delayed stomatal closure and activation of defense response repression of ABA synthesis enrichment of genes involved in AUX, CK and ET metabolism/signaling in ‘Cranbrook’	lower yield reduction in ‘Cranbrook’	[105]
transgenic *Oryza sativa* cotton	soil pot experiment in greenhouse and growth chamber irrigation reduction RWC—not performed	increased CK level modifications of source/sink relationships delayed senescence increased expression of BR-related genes	improved grain yield and grain quality improved photosynthesis, biomass accumulation	[106,107]

## 5. Conclusions

Drought is the most frequent abiotic stress adversely affecting productivity of crop plants. As sessile organisms, plants have developed sophisticated regulatory mechanisms at molecular and physiological levels to cope with water scarcity conditions. These mechanisms are important for stress survival (*biological resistance*). However, activation of these mechanisms frequently does not prevent the negative effect of drought on growth and yields (*agricultural resistance*), which is important for plant users. Therefore, there is a need for continuous and extensive research expanding the knowledge required in breeding drought-resistant and high-yielding crop genotypes. The modern breeding technologies and biotechnological approaches aimed at developing crops resilient to drought with a high yield capacity should include genomic, molecular, and physiological research. It is crucial to determine the relationships between molecular, metabolic, and physiological changes involved in *biological resistance* to drought and *agricultural resistance* estimated using agronomic criteria (photosynthetic activity, growth traits, yield). At the molecular level, research based on marker-associated selection, genome-wide association studies, and genome selection with high throughput phenotyping are useful in identifying candidate genes and TFs effective for improving the resistance of crops to drought [3,28,104,112]. Currently used approaches to obtain drought resistance crops include the use of: (a) traditional breeding programs; (b) genetically modified plants; and (c) clustered regularly interspaced short palindromic repeats (CRISPR/Cas) editing strategy [4,113]. Presently, the new strategy with possible future application is the selection of epigenetic phenotypes with increased drought resistance [71].

A slightly different, non-genetic, approach for improving crop resistance to drought is the exogenous application of natural substances, including plant metabolites ([113] and references therein). The favorable effect of such metabolites on *biological resistance* has been demonstrated. It is reported in the literature that many of these metabolites are also involved in crop yield improvement under drought (*agricultural resistance*). It is a strategy that is easy and feasible to implement. However, the beneficial effects of application of this strategy depend on the concentration of the used compound, time of application, and crop species. The use of these metabolites by producers should be preceded by long-term experiments under field conditions in order to evaluate the dose, method, and time of application in different plant species as well as the cost of application in the field.

## Figures and Tables

**Figure 1 plants-11-00922-f001:**
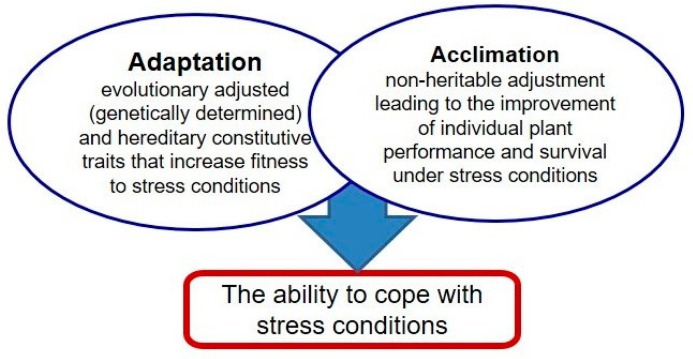
Stress adjustment developed in plants.

**Figure 2 plants-11-00922-f002:**
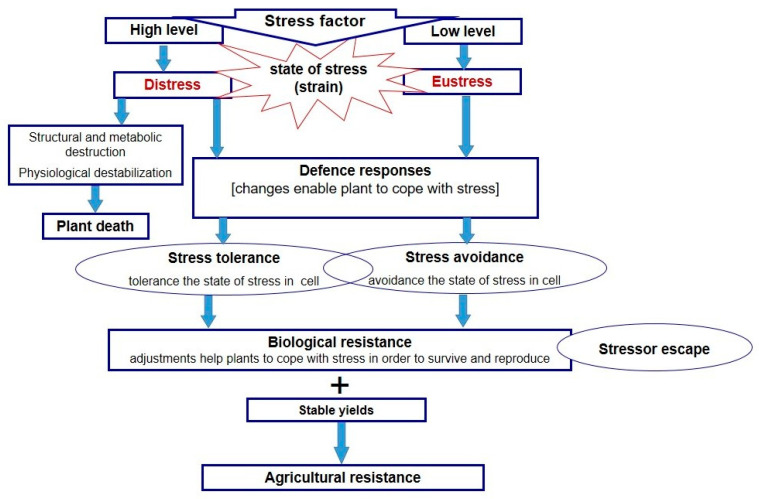
Plant responses to abiotic stress factors, coping strategy, and resistance.

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
