# Peer review of "Drought Stress Responses: Coping Strategy and Resistance"

_plants, 2022, doi:10.3390/plants11070922_

Round 1

Reviewer 1 Report

Authors address all my previous concerns.

Reviewer 2 Report

The author has addressed most of my concerns raised in my previous comments. The choice to focus only on drought stress is relevant, as it simplifies the key messages and avoids getting lost in the vast category of abiotic stresses.

Now the manuscript is improved and clearer to me and I hope it will be useful to the plant community.

This manuscript is a resubmission of an earlier submission. The following is a list of the peer review reports and author responses from that submission.

Round 1

Reviewer 1 Report

The manuscript is well-written and has a correct structure. It sumarizes several observations of the plant stress field but it focus mainly on drought stress which is not take in consideration all other abiotic stresses. So my first reccomendation is to change the title of the article. Later, the author focus on the concepts of biological resistance and agricultural resistance but I do not observe much more examples of agricultural resistance. Moreover, the manuscript figures look similar, trying to explain the different concepts but these are not much informative than reading the manuscript. Maybe a scheme, or photograph will help to understand these concepts to readers. Also, the inclusion of a table with examples of crops that meet the agricultural resistance concept will be of help. I think these changes will improve the manuscript.

Reviewer 2 Report

In this manuscript, the author reviews the theme of abiotic stress responses by describing the strategy and resistance of plants. Although this theme is rich and dynamic in plant science, there are few recent reviews on it, so this work is particularly valuable. This document is well written, and I appreciate the effort of vulgarization by schema. I still have some concerns which, in my opinion, deserve to be considered by the authors in order to gain clarity and impact.

First, I am very surprised that this review, on such an important and dynamic topic, is based on only 62 references in total, of which only 24 recent (less than 5 years). To help this synthesis work I propose some ideas of concept to detail which in my opinion are missing :

i) the isohydric and anisohydric concepts. Depending on how narrowly they control water potential, plants are classified as either isohydric (strict stomatal control, resulting in a minimum, threshold water potential for stomatal closure) or anisohydric (less tight control, with no discernible threshold) as described in:

  • Maseda, P. H., & Fernández, R. J. (2006). Stay wet or else: three ways in which plants can adjust hydraulically to their environment. Journal of experimental botany, 57(15), 3963-3977.
  • Tardieu, F., & Simonneau, T. (1998). Variability among species of stomatal control under fluctuating soil water status and evaporative demand: modelling isohydric and anisohydric behaviours. Journal of experimental botany, 419-432.

ii) the epigenomics mechanisms. Plant responses to abiotic stresses can be transient to provide plants with the required tools to acclimate and survive, whereas others may promote a state that we will refer to as “memory” which predisposes the plant for a more efficient stress response upon next encounter of stress. This some recent review in this subject:

  • Mozgova, I., Mikulski, P., Pecinka, A., & Farrona, S. (2019). Epigenetic mechanisms of abiotic stress response and memory in plants. Epigenetics in plants of agronomic importance: fundamentals and applications, 1-64.
  • Kakoulidou, I.; Avramidou, E.V.; Baránek, M.; Brunel-Muguet, S.; Farrona, S.; Johannes, F.; Kaiserli, E.; Lieberman-Lazarovich, M.; Martinelli, F.; Mladenov, V.; Testillano, P.S.; Vassileva, V.; Maury, S. Epigenetics for Crop Improvement in Times of Global Change. Biology 2021, 10, 766. https://doi.org/10.3390/biology10080766
  • Sun, C.; Ali, K.; Yan, K.; Fiaz, S.; Dormatey, R.; Bi, Z.; Bai, J. Exploration of Epigenetics for Improvement of Drought and Other Stress Resistance in Crops: A Review. Plants 2021, 10, 1226. https://doi.org/10.3390/plants10061226
  • Miryeganeh M. Plants’ Epigenetic Mechanisms and Abiotic Stress. Genes. 2021; 12(8):1106. https://doi.org/10.3390/genes12081106

iii) Antioxidants and Reactive oxygen species (ROS) homeostasis. We know that ROS has a dual function in abiotic stresses where, the same molecule can be toxic to cells at high levels, but be a signal transducer function that activates a local defense response in plants against stress. A large literature exists to understand the different plant strategies of the ROS scavenging mechanisms or antioxidants production and could be mentioned in this review. This is some examples:

  • Gómez, R., Vicino, P., Carrillo, N., & Lodeyro, A. F. (2019). Manipulation of oxidative stress responses as a strategy to generate stress-tolerant crops. From damage to signaling to tolerance. Critical reviews in biotechnology, 39(5), 693-708.
  • Magaña Ugarte, R., Escudero, A., & Gavilán, R. G. (2019). Metabolic and physiological responses of Mediterranean high‐mountain and alpine plants to combined abiotic stresses. Physiologia plantarum, 165(2), 403-412.
  • Bernardo, S., Dinis, LT., Machado, N. et al. Grapevine abiotic stress assessment and search for sustainable adaptation strategies in Mediterranean-like climates. A review. Sustain. Dev. 38, 66 (2018). https://doi.org/10.1007/s13593-018-0544-0
  • You, J., & Chan, Z. (2015). ROS regulation during abiotic stress responses in crop plants. Frontiers in plant science, 6, 1092.
  • Nadarajah, K.K. ROS Homeostasis in Abiotic Stress Tolerance in Plants. J. Mol. Sci. 2020, 21, 5208. https://doi.org/10.3390/ijms21155208

I also admit that I am a little frustrated by the lack of discussion around these concepts of resistance to stress and agriculture, which seems essential to me. There is a very large recent literature on this subject which would have its place here.

  • Raza A, Razzaq A, Mehmood SS, Zou X, Zhang X, Lv Y, Xu J. Impact of Climate Change on Crops Adaptation and Strategies to Tackle Its Outcome: A Review. Plants. 2019; 8(2):34. https://doi.org/10.3390/plants8020034
  • Dresselhaus T, Hückelhoven R. Biotic and Abiotic Stress Responses in Crop Plants. Agronomy. 2018; 8(11):267. https://doi.org/10.3390/agronomy8110267
  • Tardieu, F., Simonneau, T., & Muller, B. (2018). The physiological basis of drought tolerance in crop plants: a scenario-dependent probabilistic approach. Annual review of plant biology, 69, 733-759.
  • Godoy, F., Olivos-Hernández, K., Stange, C., & Handford, M. (2021). Abiotic Stress in Crop Species: Improving Tolerance by Applying Plant Metabolites. Plants, 10(2), 186.
  • Gilliham, M., Able, J. A., & Roy, S. J. (2017). Translating knowledge about abiotic stress tolerance to breeding programmes. The Plant Journal, 90(5), 898-917.
  • Sharma I, Kaur N and Pati PK (2017) Brassinosteroids: A Promising Option in Deciphering Remedial Strategies for Abiotic Stress Tolerance in Rice. Front. Plant Sci. 8:2151. doi: 10.3389/fpls.2017.02151
  • Hinojosa, L., González, J. A., Barrios-Masias, F. H., Fuentes, F., & Murphy, K. M. (2018). Quinoa abiotic stress responses: A review. Plants, 7(4), 106.

The author can also rely on these different works:

  • Chaudhry, S., Sidhu, G.P.S. Climate change regulated abiotic stress mechanisms in plants: a comprehensive review. Plant Cell Rep (2021). https://doi.org/10.1007/s00299-021-02759-5
  • Volaire, F. (2018). A unified framework of plant adaptive strategies to drought: crossing scales and disciplines. Global change biology, 24(7), 2929-2938. https://doi.org/10.1111/gcb.14062
  • Zarattini, M., & Forlani, G. (2017). Toward unveiling the mechanisms for transcriptional regulation of proline biosynthesis in the plant cell response to biotic and abiotic stress conditions. Frontiers in plant science, 8, 927.

Reviewer 3 Report

This manuscript reviews the coping strategy in plants to stresses. In particular, the author used drought as an example to discuss the different mechanisms used in plants to cope with this stress. I could see the author had given some thinking to this subject, but my general impression to the paper is its lack of a clear logic. I appreciate the author's effort in trying to make this a review with some novelty, given the popularity of the subject. I am wondering whether the author can add a "Conclusion" part to simply summarize the "new points" in his review when compared to those published ones. In addition, I have a few more comments for the author to consider:

1) multiple grammar issue; for example, i) the first sentence in the Abstract is problematic; ii) line 14, what is ikikresistance?; iii) line 15 also has grammar issue. I just listed these a few. There are many more in the paper.

2) an internet link should not be used; see line 25

3) A few revision marks are still in the manuscript. They need to be removed.

4) Many paragraphs are too long. For example, Line 196 to 252. Please shorten them or break up them to more than one paragraphs. These long paragraphs are very hard to follow. I had to read a few times before I know the main points. 

5) The review on the physiological part is good, but it lacks information in the molecular part.